# Interactions of construal levels on programming ability and learning satisfaction: A case study of an Arduino course for junior high school students

**Huan-Ming Chuang**[1], **Chia-Cheng Lee**[2]*

**1** Department of Information Management, National Yunlin University of Science and Technology, Douliu, Yunlin, Taiwan, **2** Department of Information Management, National Yunlin University of Science and Technology, Douliu, Yunlin, Taiwan

* d10523003@yuntech.edu.tw

**Data Availability Statement:** All relevant data are within the manuscript and its Supporting Information files.

## Abstract

Programming is one of the most crucial abilities for students in science and technology courses. Few studies on programming ability have considered the effect of students' construal levels on their learning performance. Therefore, the effects of students' construal level were explored in this study to fill this research gap and open a new avenue for the improvements in programming ability. The research participants were 110 seventh- and eighth-grade students with basic programming abilities taking an Arduino course. Data were collected from online questionnaires and analyzed using two-way analysis of variance and structural equation modeling to investigate the relationships among construal levels, programming ability, and learning satisfaction. The results revealed that students' construal levels affect their learning satisfaction and programming ability. These findings indicate that teaching strategies could effectively improve the learning satisfaction and programming ability of junior high school students.

## Introduction

In 2019, Taiwan implemented a 12-year national education curriculum. In this curriculum, students in grades seven to nine must attend two technology courses weekly, namely information technology (IT) and life technology (LT), with IT changing from an elective course to a compulsory course. Therefore, the planning and designing of appropriate teaching materials and methods are crucial in this field. The Information Technology Curriculum Manual, published in 2016, indicated that an IT curriculum should be focused on computational-thinking to cultivate aspects such as logical thinking and systematic thinking through the acquisition of computer-science–related knowledge. IT course design emphasizes improvement in students'-computational-thinking, problem-solving, teamwork, and innovative-thinking abilities. The primary topics in the seventh- and eighth-grade curricula are algorithms and programming;

**Funding:** The authors received no specific funding for this work.

**Competing interests:** The authors have declared that no competing interests exist.

therefore, this study treated students'programming ability (PA) as their essential IT skills to investigate approaches for improvements in science and technology education.

Studies related to PA frequently concentrate on specific programming languages. For instance, Palumbo and Reed [1] investigated the effect of BASIC programing language on problem-solving ability. Other studies emphasized the desirable conditions for facilitating the learning of programming skills, such as problem-based game projects [2] and playful learning environments [3]. Others still stressed the importance of computational-thinking; for instance, Brennan and Resnick [4] used the Scratch context to develop young people's computational concepts, practices, and perspectives. Although these studies have contributed to leveraging the power of PA, other effective approaches still merit exploration.

Chen and Tu [5] investigated the effect of learning attitudes and problems on learning outcomes among high school students in computer courses. They determined that learning attitudes were positively correlated to satisfaction. In other words, the students' cognitive-level and behavioral-level attitudes affected their satisfaction and degree of self-efficacy in computer-related subjects. Therefore, learning satisfaction (LS) is essential for the motivation to achieve better learning performance.

Construal level theory (CLT) [6] suggests that psychological distance, which is a subjective experience of the closeness of something from the self at a given moment, affects individuals' conceptualization and implementation of a specific plan. Accordingly, CLT can be expected to exert significant moderating effects on principal relationships. For instance, Ho, Ke, and Liu [7] identified that a higher construal level (CL) strengthened perceived ease of use more than perceived usefulness did when deciding whether to adopt a new e-learning system. Similarly, Kim, Sung, Lee, Choi, and Sung [8] recognized that individuals responded more favorably to abstractly framed desirability-focused messages posted on their Facebook news feed, whereas they demonstrated a more positive reaction to concretely framed feasibility-focused messages posted on their Facebook timeline page. Research on CLT has yet to consider the influence of students' CLs on their PA and LS.

The new science and technology curricula should motivate students to learn as well as emphasize programming and applications. Therefore, students' internal motivation and ability should be considered. Therefore, this study explores the application of CLT to enhance LS and PA as well as the core competencies of science and technology courses. The major objectives of this study can be summarized as follows: (1) investigate the relationship between PA and LS, (2) investigate the relationship between CLs and LS, (3) investigate the moderating effect of CLs on the relationship between PA and LS, and (4) suggest strategies for the planning and design of science and technology curricula.

## Literature review

**Learning satisfaction.** Elia, Solazzo, Lorenzo, and Passiante [9] mentioned LS as the "perception of enjoyment and accomplishment that learners develop in learning environments." Similarly, Kuo, Walker, Belland, Schroder, and Kuo [10] defined LS as "student perceptions of the extent to which their learning experiences were helpful and enjoyable." Deci, Ryan, and Williams [11] conceptualized satisfaction as a spontaneous experience associated with intrinsic motivation and fully internalized extrinsic motivation that meets the basic psychological needs for autonomy, competence, and relatedness. LS results in both a higher commitment to learning programs and demand to conclude the course [9, 12].

Topala and Tomozii [13] argued that the majority of studies on satisfaction in learning have referred either to a broader, general feeling regarding the overall process or a specific multifaceted context. Studies that have adopted the multifaceted approach have indicated that LS is

influenced by factors such as course content, location and facilities, teacher's teaching skills and individual characteristics, and students' participation [14–16]. Furthermore, Verkuyten and Thijs [17] emphasized the positive effect of academic and social climates in the class on students' LS levels. Therefore, we can assume that pedagogical and social factors in the classroom affect students' LS. For example, Kangas, Siklander, Randolph, and Ruokamo [3] asserted that students' LS was determined primarily by their satisfaction with the course, overall satisfaction with schooling, and satisfaction with the teacher.

**Cooperative learning.** Cooperative learning refers to a teaching method that divides students into small groups to encourage interaction and enhance students' learning performance and interpersonal behaviors [18–20]. On the basis of the definitions of cooperative learning, the following features of this teaching strategy can be identified: (1) cooperative learning is a systematic teaching strategy, (2) cooperative learning involves study groups of more than two people, (3) the members of these groups have common learning goals, (4) discussions can be held within these groups, and (5) cooperative learning can facilitate students' cognitive, social, and emotional development, thereby further promoting group learning.

Previous studies have explored the significant benefits of cooperative learning. Munir, Baroutian, Young, and Carter [21] highlighted that cooperative learning offers numerous benefits to students and teachers, including improvement of students' deep-learning and critical-thinking abilities. Parker [22] indicated that the heterogeneous groups in cooperative learning provide an environment that enables students to learn alongside their peers, support one another, offer constructive criticism, share their views, and share their results. Additionally, Johnson and Johnson [19] reported that cooperative learning encourages face-to-face interaction for problem solving, exchange of ideas, and mutual assistance. Nattiv [23] asserted that cooperative learning enables students to work collaboratively in small groups to achieve common goals; furthermore, each group member is responsible for their own learning and is interdependent in terms of rewards, work, materials, and roles. Finally, Ishler, Johnson, and Johnson [24] argued that compared with competitive and individual effort, cooperative effort results in higher performance and productivity, more positive and supportive relationships, and more improved psychological health and wellbeing.

**Construal level theory.** Liberman, Trop, and Stephan [25] defined psychologically distant events as those not present in the direct experience of reality, the four major dimensions of which being temporal space, physical space, social space, and hypotheticality. They further proposed CLT and claimed that an individual's direct experience of present reality was the starting point on which psychological distance was anchored and that any other differences were related to mental construal. CLT proposes that psychological distance changes people's responses to future events by altering their mental construal of those events. The greater the psychological distance, the more likely events are to be construed in terms of more abstract and central features (high CL) rather than in terms of more specific and incidental details (low CL). As demonstrated by Liberman et al. [25], CLT reveals that people have more abstract interpretations of psychologically distant events than of psychologically close ones.

CLT affects people's approach to decision making. For example, Vered and Nira [26] reported that when making decisions, people seek relevant information for guidance. Decision makers with higher CLs generally gather more information to ensure that they can visualize multiple outcomes before making a decision. Schwartza, Eyalb, and Tamir [27] indicated that guiding people in increasing their CLs may enhance their appreciation for the broader and more goal-relevant implications of their choices and therefore enhance their ability for self-control. People with higher CLs are reportedly more likely to consider the potential instrumentality of emotions. CLT is widely applied in different fields. For example, Kim et al. [8] investigated the effectiveness of advertising messages according to different CLs and framings.

Furthermore, in a study on participants' attitudes toward using a new e-learning system, Ho et al. [7] reported that a higher CL strengthened the effect of perceived ease of use but mitigated the effect of perceived usefulness.

According to CLT, in the context of learning in school, students' CLs related to goal-directed activities can be expected to change with increases in temporal space, as displayed in Fig 1.

Liberman and Trop [28] proposed that in goal-directed activities, the desirability of the activity's final state indicates high-level construal, whereas the feasibility of attaining this final state indicates low-level construal. The results of four related studies revealed that distant future activities were construed on a higher level than near-future activities; decisions regarding distant future activities were more influenced by the desirability of the final state and less influenced by the feasibility of attaining the final state. Furthermore, when students were confronted with a choice of academic assignments with different degrees of difficulty (feasibility) and appeal (desirability), students were more concerned with the assignment's appeal when choosing a distant future assignment, whereas they were more concerned with the difficulty when choosing a near-future assignment.

**Scratch and Arduino.** According to the curriculum outline for science and technology, the principal learning content for seventh- and eighth-grade IT curricula is algorithms and programming, indicating that programming-related courses are essential at the secondary school stage. Topalli and Cagiltay [13] evaluated students enrolled in a fourth-grade introductory programming course and determined that using Scratch, a real game development program, improved students' performance in graduate programs. This finding revealed that Scratch lays the foundation for the improvement of engineering students' PA.

Scratch is a programming language developed specifically for children by the MIT Media Lab. Scratch can improve children's computational-thinking abilities and has been extensively used for this purpose [4, 29]. Scratch language converts language commands into building blocks and interfaces from the command line to dynamic icons; therefore, it is user friendly, regardless of a user's age, background, and interests. Thus, Scratch enables users to create projects, such as interactive stories, games, animations, and simulations. A Scratch project consists of a set of roles for which behaviors may be defined using language commands and then enacted on stage. These commands can be personalized by uploading features such as photos, voice excerpts, and music clips to the "Environment" website [29] to share with others or reuse.

Arduino refers to an open-source electronics platform and the software used to manipulate it. Arduino can be controlled by proper programming for easy-to-use hands-on operation.

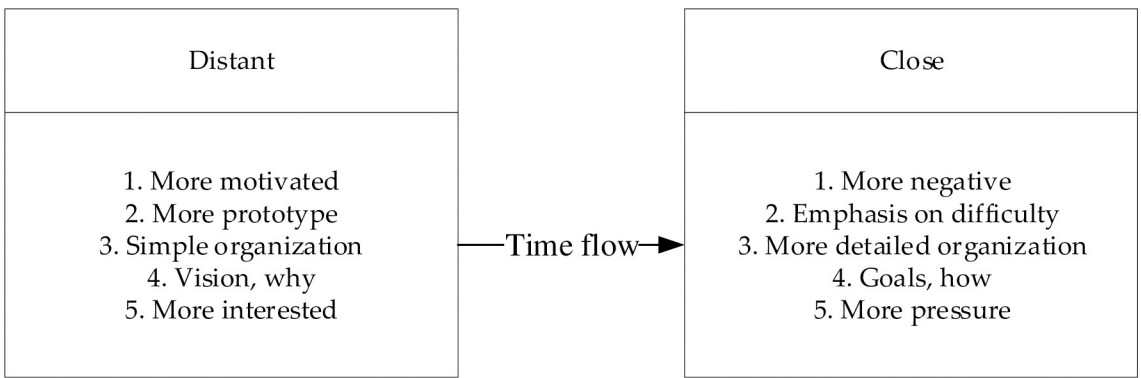

**Fig 1. Changes in construal level during goal-oriented activities.**

Arduino boards can accept inputs (e.g., light on a sensor, a finger on a button, or a Twitter message) and process them into outputs (e.g., activating a motor, turning on a light emitting diode, publishing something online). Scratch and Arduino both strive for simplicity to benefit cross-domain talents. Therefore, a team in Spain combined Arduino with Scratch and launched Scratch for Arduino, which allows people to control Arduino through a graphical interface with Scratch. This approach is followed in the present study to teach participants how to use Scratch and Arduino in sequence.

## Model construction and development of hypotheses

On the basis of a literature review, this study established a research model, as displayed in Fig 2, and developed hypotheses.

Bandura's [30] theory of self-efficacy has been widely applied in behavior research in various contexts. For example, DeWitz and Walsh [31] verified that self-efficacy was significantly associated with college satisfaction. Furthermore, Caprara, Barbaranelli, Steca, and Malone [32] determined that a teachers' self-efficacy beliefs positively affected their job satisfaction and students' academic achievement [33]. Students' PA related to their self-efficacy, which inspired the following hypothesis.

**Ha**: The programming ability of students is positively related to their learning satisfaction in science and technology courses.

According to CLT, students with a high CL are more interested in abstract concepts such as "vision" and "why" aspects, whereas students with a low CL are more interested in specific concepts, such as "goal" and "how" aspects. Liberman and Trope [28] indicated that students are more interested in homework when choosing distant future assignments, whereas they pay more attention to the difficulty when choosing near-future assignments. Furthermore, Garcia [34] conducted an experiment in which undergraduate students were asked to assess their life satisfaction and LS in relation to the near or distant future. The results indicated that the students' perceived their life satisfaction as more desirable in the distant future. Therefore, perceived life satisfaction was influenced by temporal distance, in line with CLT. Furthermore, CLT extends the notion of temporal distance to psychological distance [35, 36]. Thus, students who have higher CLT with wider psychological distance can be expected to also have higher LS. Therefore, we present the following hypothesis.

**Hb**: The construal levels of students are positively related to their learning satisfaction in science and technology courses.

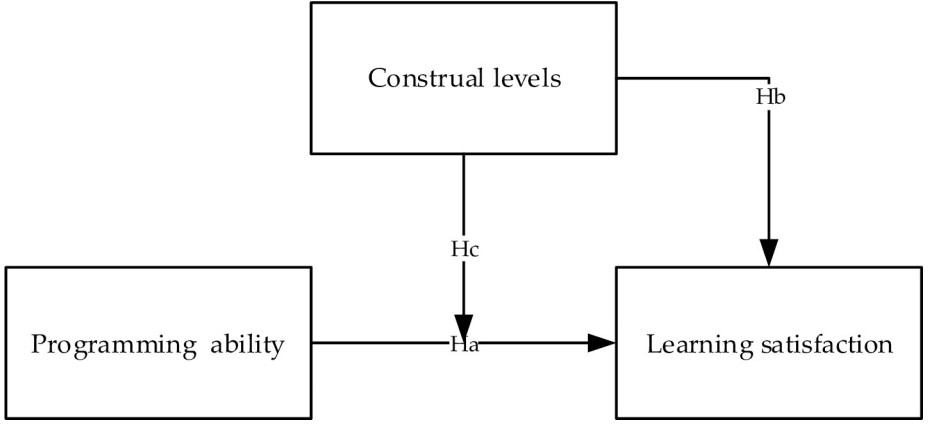

**Fig 2. Research model and hypothesized relationships.**

CLT stipulates that in the process of learning, students with lower CLs focus more on the feasibility of learning objectives. By contrast, students with higher CLs focus more on imagining the achievement of the learning objectives. A study on the adoption of a new e-learning system [7] indicated that users with higher CLs paid closer attention to perceived ease of use and paid less attention to perceived usefulness. Moreover, Kim, Chung, and Lee [37] identified that consumer's response to time restrictions varied according to CL. Furthermore, Chang and Chang [38] observed a strong correlation between student motivation and LS, as was suggested by Keller [39]. Therefore, we posit the following hypothesis.

**Hc**: The construal levels of students moderate the relationship between their programming ability and learning satisfaction in science and technology courses.

## Research method

This research was done by the author (a teacher) at the teaching site based on his own findings, without affecting the teaching progress and referring to relevant literature. Therefore, we do not seek approval from the Ethics Committee.

**Sample.** This study was conducted at a junior high school located in central Taiwan; 110 students in seventh and eighth grade participated, and valid data were obtained for 95 of these students. At this school, an IT class was held once a week in a computer classroom; each student had access to a personal computer. Each participating student completed six Scratch courses before taking the Arduino course, and each student's PA was assessed according to test results and general evaluation.

**Operational definitions of research constructs.** *Learning satisfaction*. This study defined LS as the pleasure gained from learning activities and the empirical outcomes of these activities. Studies [3] have reported that essential factors leading to LS include the learning content, location and facilities, teacher's teaching skills and individual characteristics, and students' participation.

*Programming ability*. This study adopted the Scratch for Arduino approach. Before participating in this study, each student completed six Scratch programming courses that collectively constituted an introductory programming course. This study assessed students' performance in Scratch according to their PA.

*Construal level*. This study measured participants' psychological distance from the Arduino course as their CL. A larger psychological distance from events results in a higher likelihood to conceptualize objects in an abstract manner (higher level) rather than in a specific manner (low level) [25, 36]. High-level goals are related to abstraction, and thus the "why" aspect of an activity is associated with a high CL. Low-level goals are related to specificity, and thus the "how" aspect of the activity in question is associated with a low CL. Therefore, participants' preference of the "why" or "how" aspects of the Arduino course indicated their CL.

**Procedure.** The primary purpose of this study was to investigate how students' CL and PA affect their LS on an Arduino course. To test our hypotheses, a 2 (CL) × 2 (PA) experimental design was employed. Fig 3 details the stages of the experiment.

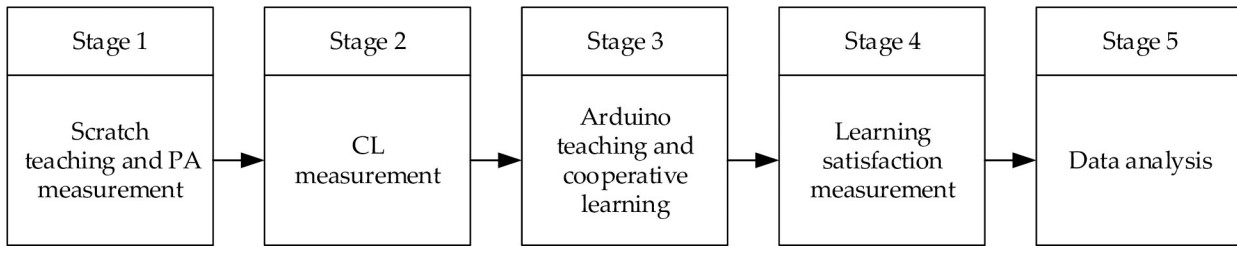

**Fig 3. Research stages.**

*Stage 1*. Stage 1 was preparatory work before the commencement of the Arduino course. The participants of this study enrolled in a 6-week (one class per week) Scratch course as part of their IT curriculum. The learning objectives were a basic understanding of how to use Scratch's interface, basic grammar, logical thinking, and understanding of the program control process. Because these skills are closely related to students' PA, an online quiz with 10 questions was developed to evaluate this ability. The test content used an example of a program that had been taught in the classroom. The test involved interpretations of the program, manipulation of the screen display, flow control, logical thinking, and calculations. Observations and test results revealed differences in students' achievements. The median score was used to divide students into high and low PA groups.

*Stage 2*. In stage 2, students were introduced to a video with the subject of "DIY Fire Fighting Robot using Arduino," which demonstrated the major functions and features of Arduino. This demonstration may have affected their perspective on further studying and practicing using Arduino, which may have led to different psychological distances. Students who were more psychologically distanced from Arduino were more likely to conceptualize it in an abstract manner (higher level) rather than in a specific manner (low level) [25]. Because higher-level goals were related to abstraction, the "why" aspect of an activity was associated with a high CL. By contrast, lower-level goals were related to specificity, and thus the "how" aspect of the activity in question was associated with a low CL. Thus, individuals with a higher CL favored more abstract concepts, such as the "vision" and "why" aspects of an activity, whereas individuals with a lower CL favored more specific concepts, such as the "goal" and "how" aspects of an activity. Therefore, this study followed the approach of Ho et al. [7]. The instructor first described two CL-oriented scenarios regarding a project that involved making an Arduino Bluetooth self-propelled car. Next, students' CLs were evaluated; under the guidance of the instructor, students specified their preferences in terms of "why" or "how," of "ease of use" or "usefulness," and "vision" or "process." According to their median score, students were divided into high and low CL groups.

*Stage 3*. Stage 3 involved teaching students how to use Arduino by using the CL method [20, 40, 41] and the "Learning Together" model [19]; students were randomly divided into heterogeneous groups of four or five and given a study assignment sheet. Each group had to complete a common assignment. The instructor emphasized the importance of teamwork and encouraged students to help each other as much as possible, actively assign roles and tasks, discuss, communicate, and complete tasks together. The instructor also provided incentives for good performance. The course consisted of three lessons, namely (1) Introduction to Arduino: become familiar with the development environment, (2) basic LED light control: learn to control a single light by adjusting the blinking time, and (3) advanced LED light control: learn to adjust the timing and sequence for the flashing of the two lights. These three lessons were necessary for using Arduino. After the submission of team projects, the instructor used tools such as a flowchart or mindmap to highlight the essential computational thinking in the assignment.

*Stage 4*. In stage 4, the students' LS with the Arduino course was measured using an 11-item online questionnaire, the items in which were adapted from those used by Kangas et al. [3] to evaluate students' satisfaction with a course, schooling, and teacher. The adapted survey also followed the multifaceted approach. The participants were required to answer all items. Questionnaires with incomplete answers were considered invalid.

*Stage 5*. Stage 5 consisted of data collation and analysis. Two-way analysis of variance (ANOVA) and PLS were performed to explore the interrelationships among students' CL,

PA, and LS. After data analyses, the proposed hypotheses were tested and conclusions were drawn.

## Data analysis and results

**Sample demographic.** Valid responses were collected from 95 students, among which 24 were in seventh grade and 71 were in eighth grade; 47 were boys, and 48 were girls. Regarding the PA test, the minimum score was 10 points, the maximum score was 100 points, and the average score was 57 points.

**Reliability and validity of the measurement.** The measurement analysis results are summarized in Table 1. The factor loading scores of all LS and CL items were greater than 0.5, indicating that the LS and CL constructs of this experiment had sufficient convergent validity. Regarding reliability, the questionnaire had a Cronbach's α greater than 0.8, indicating that the LS and CL constructs of this experiment had sufficient reliability.

**Two-way ANOVA.** The interrelationships among CL, PA, and LS were analyzed using two-way ANOVA; the results are displayed in Table 2. The analysis results revealed that both CL and PA had positive and significant effects on LS. The effect of the interaction between PA and CL on LS was also significant.

The interrelationships among PA, CL, and LS are illustrated in Fig 4. Students with high CLs had reasonably high LS; however, LS decreased slightly as PA increased. By contrast, for students with low CLs, LS increased with PA, as displayed in Fig 4.

**Hypothesis testing.** This study used structural equation modeling for parameter assessment and hypothesis testing of the proposed causal model. The component-based SEM (PLS-SEM) approach (i.e., SmartPLS) is more prediction-oriented than is a covariance-based SEM (CB-SEM) approach [42, 43] and is thus more appropriate for the initial exploratory stages of theory development [44]. Furthermore, PLS-SEM does not cause identification concerns due to small sample sizes. This study explored the effects of CL on science and technology courses in an experimental setting, thus SmartPLS was used for hypothesis testing.

**Table 1. Summary of the LS and CL scale.**

| Construct (Source) | Item | Mean | Standard deviation | Factor loading | Cronbach's α |
|---|---|---|---|---|---|
| Learning satisfaction | 1. Learning how to operate Arduino is very easy. | 3.87 | 1.082 | 0.767 | 0.908 |
| | 2. I am very happy to participate in this Arduino study. | 4.43 | 0.741 | 0.831 | |
| | 3. Learning about Arduino in the computer classroom is very easy. | 3.79 | 1.141 | 0.674 | |
| | 4. Because of the group learning design, I found learning easy. | 4.40 | 0.751 | 0.656 | |
| | 5. Learning how to operate Arduino is fun. | 4.32 | 0.907 | 0.828 | |
| | 6. Participating in this study on Arduino is a fun experience. | 4.46 | 0.663 | 0.561 | |
| | 7. The learning environment with manuals and computer programs is very interesting. | 4.33 | 0.855 | 0.801 | |
| | 8. The whole class is very excited to participate in this Arduino study. | 3.92 | 0.963 | 0.669 | |
| | 9. Group learning is an effective method for learning how to use Arduino. | 4.12 | 0.988 | 0.634 | |
| | 10. I learned a lot from this study on Arduino. | 4.32 | 0.784 | 0.830 | |
| | 11. I would be happy to participate in this study again. | 4.39 | 0.888 | 0.809 | |
| Construal level | 1. I am more concerned with "why" I should make a Bluetooth self-propelled car than "how" to make a Bluetooth self-propelled car. | 3.83 | 0.885 | 0.768 | 0.797 |
| | 2. For me, "achieving goals" is more important than "how I can learn easily." | 3.86 | 1.050 | 0.847 | |
| | 3. I am more concerned with whether learning is "useful" for my future studies than with how to make learning "simpler." | 4.16 | 0.767 | 0.869 | |

**Table 2. Summary of the two-way ANOVA to investigate the interrelationships between PA, CL, and LS.**

| Source | Type III sum of squares | df | The average sum of squares | Significance |
|---|---|---|---|---|
| Programming ability (PA) | 1.137 | 1 | 1.137 | 0.056* |
| Construal-level (CL) | 8.219 | 1 | 8.219 | 0.000*** |
| PA*CL | 1.368 | 1 | 1.368 | 0.037** |
| Error | 27.704 | 91 | 0.304 | |
| Total | 1742.182 | 95 | | |

*p < .1

**p < .05

***p < .01.

The path coefficient represented the strength and direction of a pair of constructs, and the hypothesis test verified their causal effects. The path coefficients and $R^2$ values revealed the degree of fit between the structural model and the actual data. The results for the relationships between the research framework and all the constructs are displayed in Fig 5. The $R^2$ value was 0.326; therefore, 32.6% of the total variance in LS was explained by this model. The analysis of the PLS-SEM revealed the following relationships among the constructs: CL ($\beta = 0.503$, $p < 0.05$) and PA ($\beta = 0.221$, $p < 0.05$) positively affected LS, whereas CL × PA ($\beta = -0.186$, $p < 0.05$) negatively affected LS. The results of the PLS-SEM analysis indicated that all proposed research hypotheses were supported. The results are summarized in Table 3.

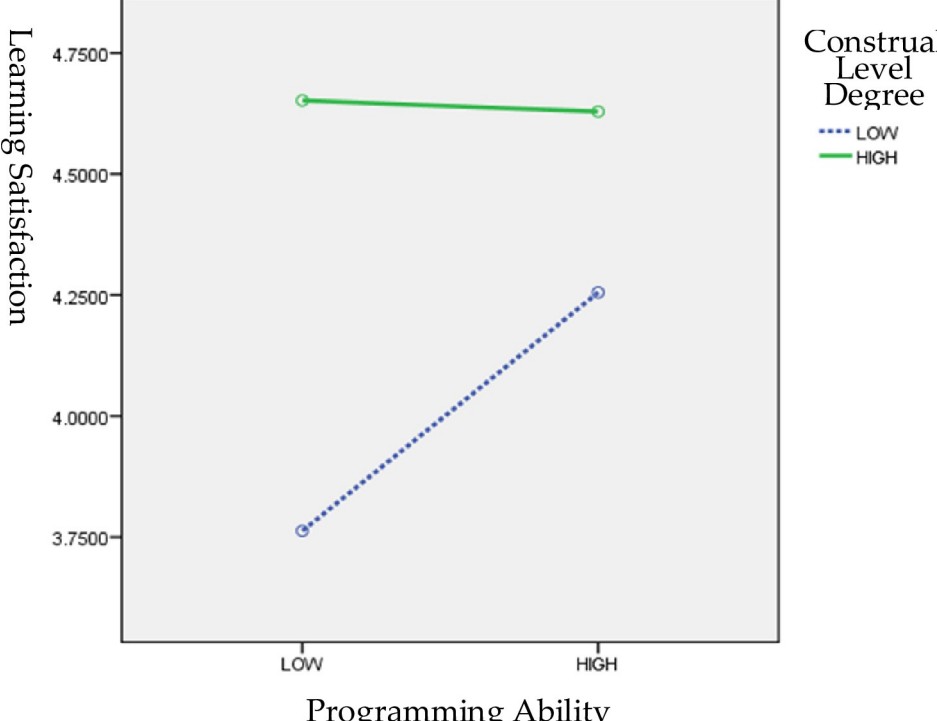

**Fig 4. Interrelationships among PA, CL, and LS.**

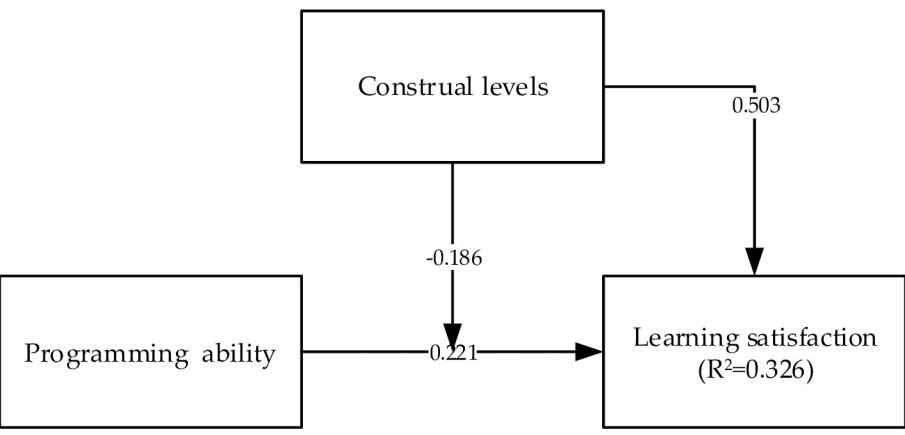

**Fig 5. Research model and PLS analysis results.**

## Conclusion and discussion

### Conclusion

Notably, students' PA and CL were determined to have positive effects on LS. However, CL had a negative moderating effect on the relationship between PA and LS.

Consequently, enhancing students' PA and CL can increase their LS. In terms of the negative moderating effect of CL, as reported in studies such as that by Lieberman et al. [25], individuals, especially those with high PA, are expected to have higher expectations for their achievement of more distant future goals. This may be because of the cold feet effect, such as when students have higher expectations for their scores on a final psychological examination at the beginning of a semester than later in the semester. Therefore, students with higher PA and CL might exhibit decreased interest and satisfaction in learning activities over time.

Furthermore, the cooperative learning method employed in this study involved elements such as "positive interdependence," "individual accountability," "face-to-face interaction," "interpersonal and small-group skills," and "group processing" [18]. Through cooperative learning, students with excellent academic performance can guide those with lower abilities. Furthermore, students were divided into groups and required to work together and rely on one another for teaching and learning purposes as well as engage in face-to-face interaction. The learning process of each group was observed, and learning achievements were assessed on the basis of each team's capacity for teamwork. Therefore, this study observed that when learning with a specific group of other students and choosing methods to complete tasks, students with higher PA and CL emphasized the creation of an environment that enabled growth to fulfill the common learning objectives of the group. However, this proved to be highly challenging.

**Table 3. Summary of the hypotheses test results.**

| Path | Standardized path coefficient | t-Value | Supported |
|---|---|---|---|
| Ha: PA → LS | 0.221 | 2.952*** | Yes |
| Hb: CL → LS | 0.503 | 7.815*** | Yes |
| Hc: PA*CL → LS | -0.186 | 2.785*** | Yes |

*** <0.01.

## Theoretical implications

This study offers several theoretical contributions. First, although CLT has been widely studied in other disciplines, such as marketing, social psychology, and information system adoption, it has not been widely applied in the teaching of programming-related courses. The findings of the present study confirmed the contingency effects of psychological factors such as CL on LS and performance. Therefore, this study opens a new avenue for improving learning performance with respect to PA in particular and science and technology courses in general.

Furthermore, this study utilized the benefits of cooperative learning and problem-based projects in science and technology courses, thus enabling students to be organized in self-managed teams and study in a playful learning environment. After achieving a team outcome, students had the opportunity to benchmark their personal performance. Furthermore, instructors used flowcharts or mindmaps to highlight computational thinking in the project. Nevertheless, this study also highlights possible adverse effects on the LS of students with high CL and PA. One approach to overcoming this problem is to create more challenging projects with specific subgoals.

## Managerial implications

With the upcoming implementation of the new 12-year national education curriculum, PA is becoming increasingly crucial in science and technology courses, and more emphasis is being placed on seeking appropriate and effective teaching content and methods in this field. The findings from this study could be applied to selecting an appropriate curriculum.

First, because PA affects students' LS, the improvement of students' PA can create a positive feedback loop in which higher PA enhances LS, course commitment, and learning performance. Second, because CL is positively associated with LS, the manipulation of CL by instructors suggests a new approach for improving LS to motivate students' learning performance regarding PA in science and technology courses. Furthermore, CL was determined to have a more significant effect than PA on LS. Therefore, priority should be given to enhancing students' CL by stressing assignments with high "goal orientation" that are "abstract" and "conceptual" in nature in science and technology curricula. Considering the influence of "psychological distance" in CLT, final teaching goals could be embedded into multiple tasks to create a sense of expectation and accomplishment at various stages of the learning process to enhance the CL of students. Third, because of the moderating effect of construal on the relationship between PA and LS, instructors of science and technology courses could assess the CL and PA of students in course planning to design customized teaching materials and methods according to students' characteristics. Furthermore, students with higher CLs tend to have higher LS than those with lower CLs; however, the negative relationship between PA and LS should be noted. Therefore, more challenging and application-oriented LT content can be incorporated into courses to maintain the LS of students with both high CL and PA.

## Limitations and directions for future research

This study had several limitations. First, science and technology encompass IT and LT, but this study focused on Arduino, which has a relatively strong focus on LT. Furthermore, because of the experimental nature of this study, the sample size was relatively small. Therefore, the generalization of the research results may be limited. Therefore, studies on other IT products with larger sample sizes are suggested to overcome this limitation. Furthermore, this study did not comprehensively consider learning models, such as mobile learning. Further research should strive to confirm the effects of usage patterns of technological products to obtain more robust results. Moreover, when students learn programming languages, the

graphical elements of such languages attract their attention. Therefore, further research could consider the effect of teaching methods and materials (graphics-oriented or text-oriented) on the interrelationships among CL, PA, and learning performance.

## Supporting information

**S1 Data.**
(PDF)

## Author Contributions

**Conceptualization:** Huan-Ming Chuang.

**Investigation:** Chia-Cheng Lee.

**Methodology:** Chia-Cheng Lee.

**Project administration:** Huan-Ming Chuang.

**Resources:** Huan-Ming Chuang, Chia-Cheng Lee.

**Supervision:** Huan-Ming Chuang.

**Validation:** Huan-Ming Chuang.

**Visualization:** Chia-Cheng Lee.

**Writing – original draft:** Chia-Cheng Lee.

**Writing – review & editing:** Huan-Ming Chuang.

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
