## [Decision Letter · Decision Letter 0]

12 Feb 2020

PONE-D-20-01854

The Effects of Construal Levels on Programming Language Ability and Learning Satisfaction: A Case Study of an Arduino Course for Junior High School Students

PLOS ONE

Dear Mr. LEE,

Thank you for submitting your manuscript to PLOS ONE. After careful consideration, we feel that it has merit but does not fully meet PLOS ONE’s publication criteria as it currently stands. Therefore, we invite you to submit a revised version of the manuscript that addresses the points raised during the review process.

We would appreciate receiving your revised manuscript by Mar 28 2020 11:59PM. To enhance the reproducibility of your results, we recommend that if applicable you deposit your laboratory protocols in protocols.io, where a protocol can be assigned its own identifier (DOI) such that it can be cited independently in the future. For instructions see: http://journals.plos.org/plosone/s/submission-guidelines#loc-laboratory-protocols

We look forward to receiving your revised manuscript.

Kind regards,

Haoran Xie

Academic Editor

PLOS ONE

Journal Requirements:

1. Please clarify whether students were required to fill out a questionnaire for the purpose as your study or as part of regular teaching procedures. If the former, then please confirm with your Institutional Review Board that prospective review and approval was not necessary and add this correspondence as "Other" file.

2. If materials, methods, and protocols are well established, authors may cite articles where those protocols are described in detail, but the submission should include sufficient information to be understood independent of these references (https://journals.plos.org/plosone/s/submission-guidelines#loc-materials-and-methods). In order to improve replicability and reproducibility, please provide supporting materials enabling other teachers and researchers to replicate your teaching intervention such as sample worksheets, a detailed lesson plan or curriculum or other such educational materials. If you include supporting materials, they should not be under a copyright more restrictive than CC-BY.

Additional Editor Comments (if provided):

The paper should be carefully revised according to the review comments from the reviewers. In addition, the authors are suggested to seek a professional English editing service for proofreading the manuscript.

Reviewers' comments:

Reviewer's Responses to Questions

**Comments to the Author**

1. Is the manuscript technically sound, and do the data support the conclusions?

Reviewer #1: Yes

Reviewer #2: Partly

2. Has the statistical analysis been performed appropriately and rigorously? 

Reviewer #1: Yes

Reviewer #2: Yes

3. Have the authors made all data underlying the findings in their manuscript fully available?

Reviewer #1: Yes

Reviewer #2: Yes

4. Is the manuscript presented in an intelligible fashion and written in standard English?

Reviewer #1: Yes

Reviewer #2: No

5. Review Comments to the Author

Reviewer #1: To investigate the relationships among students’ construal levels, programming language ability, and learning satisfaction, researchers in this study have collected related data from 95 grade seven and eight students who have taken the Arduino programming course. Two-way ANOVA and structural equation were used to analyse the data. The results revealed that construal level moderates the relationship between programming language ability and learning satisfaction negatively. Researchers proposed some strategies to improve students learning accordingly. Generally, this paper is well-structured and has valid experimental foundations. However, the authors may want to improve the work from the following perspectives.

1) The abstract has 183 words, it is relatively lengthy. The language in this session is not concise enough. Some important information that the audience expects to know cannot be found in the abstract, like the sampling number and the questionnaires. While some less significant information, for example, the features of Arduino, could be introduced in somewhere else to make the abstract more effective.

2) Chinglish is a crucial issue in the introduction section. Using “the Education Bureau of Taiwan”, “for two technology courses weekly” et al., can make the language more readable.

3) In fact, some other language errors can be found throughout the manuscript, the review suggests the authors to carefully proofread the article.

4) The references in the literature review part need to be strengthened. For example, there should be citations for the claim “Assessment of students’ LS from multiple perspectives (students’ LS concerning PLA, their schools, and their teachers) revealed…”.

5) Construal Level Theory has been introduced both in the introduction part and literature review part with considerable space, the reviewer suggests combining the two parts and highlighting the related studies.

6) It is confusing to have “Moore motivated” in Figure 1.

7) Please define the relationship of “Arduino” and “Scratch”, if we have “Arduino” both in the article title and abstract, we expect to have the literature review section named “Arduino” instead of “Scratch”. Or the authors may think another way to represent them, just keep consistent.

8) The reviewer recommends moving the position of “Operational definitions of research constructs” section to the front to assist the audience to understand the concepts in a logical way.

9) Please keep the table font consistent.

10) The “first, second, third” point sentences in the conclusion part could be improved.

11) Need citations for the sentence “A series of studies revealed that individuals have higher expectations of long-distance performance than of short-distance performance”, and the claim of “cold feet”.

12) This article could be more insightful if the authors integrate the computational thinking element in the discussion part.

13) It is suggestible to change the position of the last two paragraphs of the article and frame the language properly.

Overall, the authors have shown their efforts to report an experimental study in the field.

Reviewer #2: This study tries to explore the effects of students’ construal levels and programming language ability on learning satisfaction. However, the current presentation and results of the manuscript may not live up to expectation. My major concerns are as follows.

Research model and research hypotheses, that drive the paper, should be built from an ongoing and pertinent bibliography (up to 2020). Identifying a research gap is not enough; key is showing its significance to the field.

The research process and methodology are not clearly presented and described, and current descriptions are quite puzzling. Several issues should be clearly stated. First, how do you measure programming language ability? What type of online test were used? Was programming language ability measured only before Arduino course? How about after taking the Arduino course? I say this cause I think that the relationship between the change in programming language ability and the learning satisfaction may be worth investigated.

In addition, it is not very clear how you measured construal levels, did you just used the three questions in Table 2? Are there any literature supporting your use of such questions as measures of construct levels? Have you also used any qualitative method? Also, as you mentioned, “According to the results, students were divided into high- and low- construal level group”, how did you defined high- and low- construal level?

As the authors mentioned about a use of cooperative learning approach for course teaching, how did you divide different groups? More detailed information should be provided.

There is something wrong with Table 4, which is expected to present SEM results, however, currently, it is the same with Table 3. Without complete statistical results, it is not possible to judge your findings. Please do carefully check your manuscript before submitting it.

Besides, as for the sample size, I would say that a sample of 95 for conducting SEM would not be very persuasive. At least you can justify that the use of a sample of 95 is enough to clarify your analysis.

The presentation and structure as well as the language use should be further improved.

6. PLOS authors have the option to publish the peer review history of their article (what does this mean?). If published, this will include your full peer review and any attached files.

Reviewer #1: No

Reviewer #2: No

---

## [Author Response · Author response to Decision Letter 0]

13 Apr 2020

Dear Mr. Xie,

Thank you for your assistance and reminders

The re-uploaded content includes the modified cover letter, a response to the reviewer, a marked copy of the manuscript, an unmarked version of the original and data set as a Supporting Information file. According to the reviewer's suggestions, we have made many revisions to the content of the article, including the format of the article.

Thanks again for your assistance

Kind regards,

CHIACHENG LEE

---

## [Decision Letter · Decision Letter 1]

11 May 2020

PONE-D-20-01854R1

Effects of Construal Levels on Programming Ability and Learning Satisfaction: A Case Study of an Arduino Course for Junior High School Students

PLOS ONE

Dear Mr. LEE,

Thank you for submitting your manuscript to PLOS ONE. After careful consideration, we feel that it has merit but does not fully meet PLOS ONE’s publication criteria as it currently stands. Therefore, we invite you to submit a revised version of the manuscript that addresses the points raised during the review process.

Please address the review comments carefully.

We would appreciate receiving your revised manuscript by Jun 25 2020 11:59PM. To enhance the reproducibility of your results, we recommend that if applicable you deposit your laboratory protocols in protocols.io, where a protocol can be assigned its own identifier (DOI) such that it can be cited independently in the future. For instructions see: http://journals.plos.org/plosone/s/submission-guidelines#loc-laboratory-protocols

We look forward to receiving your revised manuscript.

Kind regards,

Haoran Xie

Academic Editor

PLOS ONE

Reviewers' comments:

Reviewer's Responses to Questions

**Comments to the Author**

1. If the authors have adequately addressed your comments raised in a previous round of review and you feel that this manuscript is now acceptable for publication, you may indicate that here to bypass the “Comments to the Author” section, enter your conflict of interest statement in the “Confidential to Editor” section, and submit your "Accept" recommendation.

Reviewer #1: All comments have been addressed

Reviewer #2: (No Response)

2. Is the manuscript technically sound, and do the data support the conclusions?

Reviewer #1: Yes

Reviewer #2: Yes

3. Has the statistical analysis been performed appropriately and rigorously? 

Reviewer #1: Yes

Reviewer #2: Yes

4. Have the authors made all data underlying the findings in their manuscript fully available?

Reviewer #1: Yes

Reviewer #2: Yes

5. Is the manuscript presented in an intelligible fashion and written in standard English?

Reviewer #1: Yes

Reviewer #2: No

6. Review Comments to the Author

Reviewer #1: The manuscript has been improved a lot compared with the original version. I appreciate the efforts that the authors have made to make the article more publishable in the journal. However, still, there are some points that may need more attention from the authors.

1) The second sentence of the abstract makes the logic confusing. It is unrelated to the theme of the article, whether remove it or revise it.

2) The tense of the penultimate sentence of the abstract is inappropriate.

3) The third paragraph of the introduction section lacks a theme sentence at the beginning of the paragraph. In fact, the review can see only limited value of this paragraph, it is necessary for it to be there?

4) Similarly, add the theme sentences for the many paragraphs in the literature review section. Not just list different publications to compose the whole paragraph.

5) The hypothesized relationship structure indicates that this study is not investigating the “Effects of Construal Levels on Programming Ability and Learning Satisfaction”, but the “Interactions of Construal Levels on Programming Ability and Learning Satisfaction”, think about correcting the title to be consistent with the research itself.

6) Please change the size of table one to make it tidier, remove the shadow of table 2 and table 3. Address front size problems in the reference section. Not happy to see this kind of errors at this stage.

Good luck!

Reviewer #2: Most of the major concerns have been solved. There are still minor issues. First, some of the dated references in the papers could be replaced by more recent ones (including 2018-20).

Besides, presentation should be further improved. For example, please provide page number for:

Sweeney and Ingram [39] defined LS as the “perception of enjoyment and accomplishment that learners develop in learning environments.”

The same for:

Similarly, Kuo, Walker, Belland, Schroder, and Kuo [26] defined LS as “student perceptions of the extent to which their learning experiences were helpful and enjoyable.”

Also, inconsistent tense use issues, e.g., tense of “provide” and “encouraged” in the following paragraph should be consistent.

“Parker [35] indicated that the heterogeneous groups in cooperative learning provide an environment that enables students to learn alongside their peers, support one another, offer constructive criticism, share their views, and share their results. Johnson and Johnson [18] reported that cooperative learning encouraged face-to-face interaction to solve problems, offer mutual assistance, and share ideas.”

In addition, formation issue, e.g., “Figure 3 details” and “the stages of the experiment.” should be aligned.

Please double-check the whole manuscript.

7. PLOS authors have the option to publish the peer review history of their article (what does this mean?). If published, this will include your full peer review and any attached files.

Reviewer #1: No

Reviewer #2: No

---

## [Author Response · Author response to Decision Letter 1]

29 Jun 2020

Reviewer #1

The manuscript has been improved a lot compared with the original version. I appreciate the efforts that the authors have made to make the article more publishable in the journal. However, still, there are some points that may need more attention from the authors.

Thank you for your warm encouragement for our efforts. We appreciate your time, energy, and insight in reviewing our manuscript, which has led to an improved paper. We responded to your suggestions below.

1) The second sentence of the abstract makes the logic confusing. It is unrelated to the theme of the article, whether remove it or revise it.

We agree with your suggestion and removed the sentence.

2) The tense of the penultimate sentence of the abstract is inappropriate.

We corrected this mistake.

3) The third paragraph of the introduction section lacks a theme sentence at the beginning of the paragraph. In fact, the review can see only limited value of this paragraph, it is necessary for it to be there?

4) Similarly, add the theme sentences for the many paragraphs in the literature review section. Not just list different publications to compose the whole paragraph.

Indeed, there is much room for improvement in the writing of this paper; we changed some wordings and made theme sentences more significant to make the article more readable. We also asked for help from native English speakers for professional proofreading. Your care and suggestion for the novice researcher are very touching.

5) The hypothesized relationship structure indicates that this study is not investigating the “Effects of Construal Levels on Programming Ability and Learning Satisfaction”, but the “Interactions of Construal Levels on Programming Ability and Learning Satisfaction,” think about correcting the title to be consistent with the research itself.

Thank you for this excellent suggestion. We corrected the title to be consistent with the research itself.

6) Please change the size of table one to make it tidier, remove the shadow of table 2 and table 3. Address front size problems in the reference section. Not happy to see this kind of errors at this stage.

Thank you for this diligent remind, we tried our best to correct all errors.

Reviewer #2

Most of the major concerns have been solved. There are still minor issues. 

Thank you for your warm support; we appreciate the great help from you. We responded to your suggestions below.

First, some of the dated references in the papers could be replaced by more recent ones (including 2018-20).

We replaced some references with more recent ones; please refer to 

references from 2018-20 as listed in #6,8,25,37,39,41,46.

Besides, presentation should be further improved. For example, please provide page number for: Sweeney and Ingram [39] defined LS as the “perception of enjoyment and accomplishment that learners develop in learning environments.”

The same for: Similarly, Kuo, Walker, Belland, Schroder, and Kuo [26] defined LS as “student perceptions of the extent to which their learning experiences were helpful and enjoyable.”

Thank you for this industrious remind; we followed this suggestion.

The 39th reference has been changed to a newer one, and the content is on page 590.

The content of the 26th reference is on page 164.

Also, inconsistent tense use issues, e.g., tense of “provide” and “encouraged” in the following paragraph should be consistent.

“Parker [35] indicated that the heterogeneous groups in cooperative learning provide an environment that enables students to learn alongside their peers, support one another, offer constructive criticism, share their views, and share their results. Johnson and Johnson [18] reported that cooperative learning encouraged face-to-face interaction to solve problems, offer mutual assistance, and share ideas.”

In addition, formation issue, e.g., “Figure 3 details” and “the stages of the experiment.” should be aligned.

Please double-check the whole manuscript.

We double-checked the whole manuscript to correct all related issues and asked for help from native English speakers for professional proofreading.

---

## [Editor Report · Decision Letter 2]

9 Jul 2020

Interactions of Construal Levels on Programming Ability and Learning Satisfaction: A Case Study of an Arduino Course for Junior High School Students

PONE-D-20-01854R2

Dear Dr. LEE,

We’re pleased to inform you that your manuscript has been judged scientifically suitable for publication and will be formally accepted for publication once it meets all outstanding technical requirements.

Kind regards,

Haoran Xie

Academic Editor

PLOS ONE

Additional Editor Comments (optional):

The paper has been significantly revised and can be accepted now.
---

## [Editor Report · Acceptance letter]

14 Jul 2020

PONE-D-20-01854R2 

Interactions of Construal Levels on Programming Ability and Learning Satisfaction: A Case Study of an Arduino Course for Junior High School Students 

Dear Dr. Lee:

I'm pleased to inform you that your manuscript has been deemed suitable for publication in PLOS ONE. Congratulations! Your manuscript is now with our production department. 

Kind regards, 

on behalf of

Professor Haoran Xie 

Academic Editor

PLOS ONE